# Sulodexide Inhibits Arterial Contraction via the Endothelium-Dependent Nitric Oxide Pathway

**DOI:** 10.3390/jcm13082332

**Published:** 2024-04-17

**Authors:** Nadide Ors Yildirim, Alperen Kutay Yildirim, Meric Demeli Ertus, Ahmet Onur Dastan, Bilge Pehlivanoglu, Yung-Wei Chi, Sergio Gianesini, Suat Doganci, Vedat Yildirim

**Affiliations:** 1Department of Anesthesiology and Reanimation, Sincan Training and Research Hospital, Ankara 06949, Turkey; orsnadide@gmail.com; 2Department of Cardiovascular Surgery, Faculty of Medicine, Gazi University, Ankara 06560, Turkey; 3Department of Physiology, Zonguldak Bulent Ecevit University, Zonguldak 67600, Turkey; mericdml@gmail.com; 4Department of Physiology, Hacettepe University Faculty of Medicine, Ankara 06100, Turkey; ahmetdastan@hacettepe.edu.tr (A.O.D.); bilge.pehlivanoglu@gmail.com (B.P.); 5Vascular Center, University of California, Sacramento, CA 95817, USA; willychi70@gmail.com; 6Department of Surgery, Uniformed Services University of the Health Sciences, Bethesda, MD 20814, USA; sergiogianesini@gmail.com; 7Vein Vascular Clinics, Ankara 06510, Turkey; suat_doganci@yahoo.com; 8Department of Anesthesiology and Reanimation, Gulhane Training and Research Hospital, University of Health Sciences, Ankara 06010, Turkey; drvyildirim@yahoo.com

**Keywords:** sulodexide, L-NAME, Krebs–Henseleit, mammary artery, vasodilatation, arteriopathy

## Abstract

**Background/Objectives**: Sulodexide (SDX) is a drug known for restoring the glycocalyx, thereby offering endothelial protection and regulating permeability. Additionally, it has antithrombotic and anti-inflammatory properties and has shown arterial vasodilatory effects. Endothelial cells play a crucial role in maintaining homeostasis, with their dysfunction being a key contributor to loss in vasodilatory response, especially in arterial pathologies. The aim of this study was to investigate the effects of SDX on stimulated vascular tonus in human arterial samples and to assess the function of the endothelial layer as a source of nitric oxide (NO). **Methods**: A total of 16 internal mammary artery remnants from coronary artery bypass graft surgeries were dissected into endothelium-intact and endothelium-denuded groups (n = 8 each). The arterial rings were equilibrated under tension, with their basal tonus recorded before and after phenylephrine stimulation. SDX’s impact on arterial contraction was assessed through cumulative dose–response curves. NO synthase inhibitor (Nω-nitro-L-arginine methyl ester) was used to assess SDX’s vasodilatory effect over the NO pathway. **Results**: SDX application resulted in concentration-dependent vasorelaxation in both endothelium-intact and endothelium-denuded groups at certain doses. However, the inhibitory effect of SDX was more pronounced in endothelium-intact rings at higher doses compared to endothelium-denuded rings (*p* < 0.05). Similar inhibition of contraction curves was achieved for both endothelium-intact and endothelium-denuded rings after L-NAME pre-incubation, suggesting a necessity for NO-related endothelial pathways. **Conclusions**: SDX exerts a concentration-dependent inhibition on arterial contraction, emphasizing the critical role of an intact endothelium and NO-mediated pathways in this process. This underscores SDX’s potential in treating endothelial dysfunction-related pathologies.

## 1. Introduction

Endothelial cells play crucial roles in maintaining homeostasis by regulating blood flow, vessel wall permeability, and glycemic and lipid metabolism and translating hemodynamic forces into physiological or pathological biochemical messages [1]. A pivotal player in the signaling between physics and biochemical vascular phenomena is the glycocalyx (GCX), an electronically charged carbohydrate-rich layer [2]. It is proven to be essential for normal physiological action, to be involved in pathology development, and to take part in vessel tone regulation. The disruption of the endothelial lining and structure leads to intravascular inflammation, which serves as the catalyst for the advancement of atherosclerosis and thrombosis [3]. Degradation of the GCX layer is also associated with a decrease in the release of nitric oxide (NO), which is essential for the relaxing of vascular smooth muscle cells and vasodilation [4].

Vasodilation can be induced by endothelial or non-endothelial mechanisms. Endothelium-associated vasodilation is mediated by NO, which is synthesized and released from the endothelium. The NO acts on vascular smooth muscle cells, causing relaxation, the production of prostaglandin I_2_ (PGI_2_) by activating cyclooxygenase, and/or the release of endothelium-derived hyperpolarization factor [1]. Taken together, the endothelium plays an important role in vasodilation and spreading the vasodilatory responses. Disturbance in the endothelium and the GCX, which serve as a regulator and barrier, impair the vasodilatory response and block the spreading of vasodilation [2].

Endothelial dysfunction or malfunction is implicated in both arterial and venous pathologies. The most widespread arterial pathology is chronic arteriopathy, a gradual process triggered by damage to the arterial endothelium or smooth muscle layer [5]. Chronic arteriopathy encompasses a collection of pathological conditions resulting in persistent inflammatory processes, constriction of arterial walls, and diminished blood circulation. On the other hand, chronic venous diseases are associated with hypercoagulability, venous thrombosis, endothelial damage, and rheologic changes in blood flow [6]. The social, economic, and physiological toll of chronic arteriopathy and chronic venous disease is substantial yet often overlooked.

The pharmaceutical arsenal for the prophylaxis and treatment of atherothrombotic arterial disorders and venous thromboembolism consists of antiplatelet medications with either anticoagulant or antithrombotic actions, as well as lipid-lowering therapies [7]. Examples of such medications include warfarin, acenocoumarol, unfractionated heparin (UFH), low molecular weight heparin (LMWH), and dermatan sulfate [7,8]. Sulodexide (SDX), a mixture of glycosaminoglycans isolated from porcine intestinal mucosa, is commercially available in Europe, South America, and Asia [9,10]. The sulfated or non-sulfated monosaccharides in glycosaminoglycans play a variety of crucial tasks, such as controlling the activities of proteins through influencing cytokines, adhesion molecules, and chemokines, in addition to their antiproteolytic properties [11]. SDX is made up of dermatan (20%) and heparan (80%) sulfate [12]. Heparan sulfate has less impact on coagulation than UFH or LMWH, thus reducing the risk of hemorrhage. Its increased antithrombin affinity and longer half-life are related to its bioavailability [13]. Furthermore, iduronic acid and galactosamine, the principal constituents of dermatan sulfate, are found within the endothelium and the vascular walls [14]. Factor X and II are inhibited by its anticoagulant action. Although SDX’s unique anticoagulant effect takes time to become noticeable, it is less effective than UFH and LMWH’s instant anticoagulant effects [15]. Pharmacokinetic investigations showed that the drug’s distribution volume is very large, following SDX’s affinity for the endothelium’s surface rather than the plasma proteins [16].

Numerous investigations have been conducted to examine the utilization of SDX in the context of peripheral arteriopathy, metabolic disorders linked to arteriosclerosis, vascular complications arising from diabetes, and venous pathologies [3,4,17,18,19,20,21,22,23,24,25]. Cospite et al. measured the capillary filtration coefficient to assess microcirculation changes and measured a lower coefficient in the SDX-administered group, indicating a lower capillary permeability [26]. A recent investigation demonstrated the presence of the GCX on the lymphatic human endothelium, paving the way for further explanations regarding GCX potentials in capillary filtration regulation [27]. SDX also plays a favorable role in venous ulcers as it promotes vascular repair by upregulating the expression of fibroblast growth factors, in addition to its antithrombotic and anti-inflammatory effects [28]. In the SDX arterial venous Italian study (SUAVIS), the SDX group had better ulcer healing rates and shorter recovery times [29,30]. According to the study by Luzzi et al., the prevalence of post-thrombotic syndrome was lower in the SDX-treated patients than in the patients who received standard medical care in accordance with International Union of Angiology guidelines after the discontinuation of anticoagulants for deep vein thrombosis [31]. Recent investigations on SDX also revealed its blood-pressure-lowering effect in hypertensive patients [32,33]. Previous data suggested several beneficial effects, including the inhibition of oxidative stress, the modulation of growth factors, the reduction in matrix metalloproteinase (MMP) expression, and the protection of endothelial cells [34,35,36,37,38,39,40,41].

Studies conducted in animal models have evaluated the effects of SDX on the rat aorta, mesenteric artery, and inferior vena cava, demonstrating improved vein function and venous contractility by causing a decrease in MMP levels, in addition to its antiplatelet, anti-inflammatory, and antiproteolytic benefits [18]. Moreover, SDX caused arterial relaxation via endothelium-dependent NO production [17]. We previously showed the veno-contractile effect of SDX on the human saphenous vein and its mediation by the NO synthase pathway [42]. However, functional research on the effects of SDX on human arterial tissue, which is a target of treatment in the therapeutic indications of SDX, is lacking. This study intended to investigate the effect of SDX on the stimulated vascular tonus of human arterial samples and the function of the endothelial layer as an NO source.

## 2. Materials and Methods

Patients (n = 16, 14 males and 2 females) who underwent scheduled elective double coronary artery bypass graft surgery were informed about the protocol of internal mammary artery (IMA) harvesting, and the ones who provided consent to participate were included in this study. All patients were diagnosed with coronary artery disease (CAD) and had severe stenosis (70–99%) in two of the left anterior descending, left circumflex, or right coronary arteries. Preoperative medication for the patients included metoprolol as a beta blocker, thiazide diuretics, perindopril as an angiotensin-converting enzyme inhibitor (ACE-I), and valsartan as an angiotensin receptor–neprilysin inhibitor (ARNi). Patients were not under treatment with calcium channel blockers. All medication was stopped 24 h prior to surgery. The remnants of IMA not to be used for grafting were used in this study. The protocol was approved by the Research Ethics Committee at Gulhane Faculty of Medicine, Health Sciences University (Issue 2023/304). All the researchers strictly adhered to the ethical standards of the Declaration of Helsinki.

Nitroglycerine exposure was avoided from 24 h prior to surgery until completion of IMA harvesting. The IMA segments that remained from the surgery and were considered medical waste were placed in oxygenated cold Krebs–Henseleit solution immediately and transferred to the laboratory in about 30 min. Serosa and surrounding connective tissue were dissected, and IMA segments were randomly separated into endothelium-intact and endothelium-denuded groups. Four arterial rings (~3 mm in length) were prepared from each patient and submerged in 10 mL organ baths containing Krebs–Henseleit solution (118.4 mM NaCl, 4.7 mM KCl, 1.2 mM KH_2_PO_4_, 1.2 mM MgSO_4_, 25.0 mM NaHCO_3_, 2.5 mM CaCl_2_, and 12.2 mM glucose at 37 °C, pH 7.4 and gassed with a mixture of 5% CO_2_ and 95% O_2_). The arterial rings were attached to isometric force transducers (MAY FDT2, Biopac, CA, USA) and allowed to equilibrate under 0.5–1 g tension for at least 60 min with washouts every 15 min. One of the rings was always spared as time control and not exposed to any drug but only a physiological bathing solution. The data were collected in real-time by an MP36 data acquisition and analysis system via BSLPRO software (version of 3.6.7, Biopac, Goleta, CA, USA).

After the rings were stabilized, the basal tonus was recorded for 10 min. Then, the viability of vessel rings was tested with 120 mM KCl, and non-responders (only one strip) were excluded. Following KCl-challenge for 10 min, the IMA rings were rebalanced for 60 min. Then, one of the endothelium-intact rings was pre-incubated with NO synthase inhibitor, Nω-nitro-L-arginine methyl ester (10^−4^ M, L-NAME), for 10 min. Another endothelium-intact ring from the patient was not incubated with L-NAME. Phenylephrine (PheE, 6 × 10^−7^ M) stimulation was applied to both of endothelium-intact rings for evoking arterial contraction. To evaluate the effect of SDX on arterial contraction, 10 min after PheE stimulation cumulative SDX (0.001 mg/mL, 0.005 mg/mL, 0.01 mg/mL, 0.05 mg/mL, 0.1 mg/mL, 0.5 mg/mL, and 1 mg/mL) dose–response curves were recorded. Each dose of SDX was applied in 5 min intervals without refreshing the bathing solution.

The same experimental protocols were applied to the IMA rings in endothelium-denuded groups where the endothelial layer of the vessels was mechanically destructed. After each procedure, baths were washed three times, and strips were allowed to re-equilibrate for at least an hour with washouts every 15 min. After all the protocols were completed, all the IMA rings were challenged by 120 mM KCl to affirm viability, then untied and weighed. 

The data were analyzed by an MP36 data acquisition and analysis system via BSLPRO software (version of 3.6.7) (Biopac, Goleta, CA, USA) and SPSS 23.0 statistical package for Windows 8. The force of contraction and/or tonus of the strips were normalized for tissue weight (g/100 mg of wet tissue weight) and given as the percentage of KCl-induced maximum responses. The distribution of the data was tested with the Shapiro–Wilk test. To compare drug effects within groups, repeated measures analysis of variance (ANOVA) followed by Tukey’s post hoc test for multiple comparisons was performed. Between-group comparisons were carried out by Student’s *t*-test. pEC50 values were calculated to compare the efficiency of SDX on the KCl or PheE pre-contracted arterial segments. The concentration where maximum contraction (Emax) and the negative logarithm of the concentration resulting in half the maximal relaxation (pEC50) were determined from individual concentration-relaxation curves by SigmaPlot. The number of patients/IMA segments and the prepared strips were shown by n and N, respectively. All the data are presented as mean ± standard error of mean (SEM). *p* < 0.05 (two-tailed) value was considered as statistically significant.

## 3. Results

The patients were comparable regarding demographic data, comorbidities, and previous medications between the endothelium-intact and endothelium-denuded groups (Table 1). 

Comparing the maximum force of contraction between the endothelium-intact and the endothelium-denuded control groups, the basal and KCL-induced forces of contraction were significantly higher in the endothelium-denuded group (*p* < 0.05), suggesting the absence of an inhibitory mechanism (Table 2). The difference between endothelium-intact and endothelium-denuded rings in the PheE-induced contraction force was statistically different in the control group (*p* < 0.005), pointing out the role of the endothelium in decreased vascular tone and vasodilation. 

In the L-NAME pre-incubated IMA rings, the maximum force of contraction was similar between the endothelium-intact and endothelium-denuded groups in terms of basal, KCl-induced, and PheE-induced states (Table 2). Results showed that inhibition of the vasodilatory effect was successfully achieved, and contraction occurred without any significant difference in the endothelium-intact group compared to the denuded one. There was a significant difference in terms of contraction in KCL and PheE-induced rings between the endothelium-intact control group and the endothelium-intact L-NAME pre-incubated group (^#^ *p* < 0.05). An increase in contraction force was observed in endothelium-intact L-NAME pre-incubated IMA rings, indicating the successful inhibition of the vasodilatory effect. 

The results of the cumulative SDX dose–relaxation curves for PheE-stimulated endothelium-intact and endothelium-denuded groups are given in Figure 1. SDX (0.001–1.0 mg/mL) application resulted in concentration-dependent vasorelaxation in both endothelium-intact and endothelium-denuded groups (*p* < 0.05, Figure 1A). The vasorelaxant effect became more prominent as the drug accumulated. When each IMA ring was evaluated for increasing doses of SDX, the inhibitory effect of each dose was more pronounced compared to the PheE-stimulated ring in the endothelium-intact group (*p* < 0.05). In the within-group comparison of endothelium-intact rings, the first significant relaxing effect was observed at the dose of 0.005 mg/mL SDX and became more evident with each increasing concentration. 

The inhibitory effect of SDX was also noticeable at doses of 0.05 mg/mL and higher in endothelium-denuded rings (*p* < 0.05). The curve showed a minor change, indicating a slight inhibition of contraction in endothelium-denuded rings. Nevertheless, compared to the endothelium-intact group, there was a significant difference in the inhibition of contraction at doses of 0.1 mg/mL, 0.5 mg/mL, and 1.0 mg/mL. The dose–response curve for the endothelium-denuded rings exhibited a less steep slope, indicating that the extent of final relaxation was not as pronounced as in the endothelium-intact group. The inhibition of contraction that occurred at these doses was statistically greater in the endothelium-intact group than in the endothelium-denuded group (*p* < 0.005). However, the inhibitory effect of SDX was visible in both endothelium-intact and endothelium-denuded groups at certain doses. These findings may suggest the presence of a possible non-endothelial vasodilatory mechanism affected by SDX application. 

When evaluating the effect of SDX on L-NAME pre-incubated rings, our results showed similar contractile curves for both endothelium-intact and endothelium-denuded rings (Figure 1B). Within-group analysis revealed that the vasorelaxant effect of SDX was prominent at doses of 0.05 mg/mL and higher, although the maximum relaxation observed was less in both groups. Furthermore, the response to SDX in both endothelium-denuded and L-NAME pre-incubated endothelium-intact rings was similar, highlighting the critical role of the endothelial layer in the mechanism of action of SDX on vascular contractions.

The maximum PheE-induced contraction responses of all the groups were comparable, although they failed to reach statistical significance; the highest contraction was recorded in L-NAME-exposed endothelium-intact IMA rings. Higher pEC50 values indicate that higher vascular tonus was achieved with lower doses of the stimulating agent; thus, the significantly higher pEC50 values of the L-NAME-exposed endothelium-intact rings compared to the control rings (*p* < 0.05) suggested a stronger contracting effect for PheE when the NO synthase pathway was absent. The results obtained from full-thickness arterial rings support the role of the endothelium and NO in basal and stimulated vascular tonus. (Table 3). 

The evaluation of cumulative sulodexide application and involvement of endothelium in its effect revealed similar Emax in all of the groups. However, a significantly lower pEC50 value in L-NAME pre-incubated endothelium-intact rings revealed an impaired vasorelaxant effect of SDX. In endothelium-denuded IMA rings, the relaxant effect of sulodexide on PheE precontracted rings was comparable (P control vs. L-NAME pre-incubated endothelium-denuded rings *p* > 0.05); in addition, endothelium-intact and L-NAME-exposed rings were comparable to the results of these rings (Table 3).

The endothelium-denuded control and endothelium-denuded L-NAME pre-incubated rings responded similarly, and both were more sensitive to PheE stimulation than the endothelium-intact control rings. The effect of SDX application and involvement of endothelium in contractile responses was clear in the SDX-applied groups. Regarding the results of endothelium-denuded rings to SDX, there was no difference between control and L-NAME pre-incubated conditions (*p* > 0.05). However, there was a significant decrease in the pEC50 values of both the endothelium-denuded control and the L-NAME pre-incubated groups compared to the endothelium-intact control group (*p* < 0.05). Significantly lower pEC50 values in cumulatively SDX-applied rings indicated that higher doses of SDX were required to achieve 50% inhibition of maximum contraction. These results pointed to a right shift in the SDX dose–relaxation curve in the rings where either the endothelial layer was disturbed or NO production was blocked by L-NAME.

## 4. Discussion

To the best of our knowledge, this marks the first study indicating a dose-dependent inhibitory effect of SDX in human arteries, together with an endothelial-mediated mechanism of action. The data we presented serve as progress in the research line involving rat aortas and mesenteric arteries as an animal model [17]. The major findings of this study are that SDX causes an inhibitory effect on arterial contraction in a dose-dependent manner, mainly achieved by the endothelial NO pathway.

In arteries, endothelium comprises a single layer of endothelial cells. However, smooth muscle cells are more elongated and surround the endothelium with a variable number of layers. The thickness of the muscular layer varies depending on the location, size, and function of the arteries. The vascular tonus, in other words, constriction and dilation of the arteries, is physiologically regulated. When the metabolic demand of the organs increases, or in pathological conditions such as chronic arteriopathy that result in decreased blood flow, vasodilation becomes a crucial adaptation [1]. Vasodilation can occur via endothelial or non-endothelial pathways. Endothelial mechanisms involve the release of NO from the endothelium [43], the production of PGI_2_ by cyclooxygenase activation [44], or the release of endothelium-derived hyperpolarization factor [45,46]. NO mediates a cGMP-associated decrease in intracellular Ca^2+^ and promotes vasodilation by reducing the Ca^2+^ sensitivity of contractile proteins [47]. 

SDX’s composition contributes to its antithrombotic action with a lower risk of hemorrhage, which makes it useful in the prophylaxis and management of peripheral vascular diseases. SDX also has beneficial effects on rheology and hemostasis, such as lowering the viscosity of blood and inhibiting the migration of smooth muscle cells to the innermost layer [9]. Besides these assets, SDX has the ability to manage vascular tone both over arteries and veins. A study by Rafetto et al. indicated that MMP levels increased in veins under prolonged stretch, and the contraction ability of veins was decreased [48]. A recent study by Rafetto et al. suggested that SDX improves contraction and decreases MMP-2 and -9 in veins under prolonged stretch [18]. Both this study and another previous study conducted by the same research group demonstrate a potential mechanism of endothelial NO pathway for the management of venous tonus [49]. Our previous study investigating SDX’s effect on veins showed that SDX had a dose-dependent veno-contractile effect in human saphenous veins by means of NO synthase pathways’ involvement [42]. Although SDX, with its glycosaminoglycan structure, exerts supportive effects on endothelium, such as restoring its function and regulating cytokine expression, little was known about SDX’s effect on arteries. The closest design was in an animal study by Raffetto et al. [17]. L-NAME, indomethacin, and tetraethylammonium were used for the inhibition of endothelial NO synthase [43], the PGI_2_–cAMP pathway and cyclooxygenase inhibitor [44], and endothelium-derived hyperpolarization factor [46,47], respectively, targeting all three endothelial mechanisms involved in vascular functions consecutively. It was observed that relaxation caused by SDX was abolished by endothelium denudation and NOS inhibitor. Other inhibitors did not cause a significant inhibition to SDX-induced vasodilation in rat aorta and mesenteric artery. This study was conducted in an animal model and did not involve a dose-dependent investigation of SDX. As this study was conducted in animal tissue and did not involve a dose-dependent investigation of SDX, it represents a significant step forward in the research line of SDX’s effect on vasodilation in human arteries. We investigated SDX’s arterial relaxing effect in human IMA samples, and L-NAME was used to determine the involvement of the endothelial NO synthase pathway in the inhibition of contraction. During this study, the cumulative dose–relaxation effect of SDX was also investigated (Figure 1). According to our findings, SDX significantly inhibited PheE-induced contraction in endothelium-intact arterial samples compared to endothelium-denuded samples (Figure 1A). As SDX’s dose accumulated and reached 0.1 mg/mL, inhibition of contraction became more distinct between the two groups (Figure 1A). In the endothelium-intact IMA samples, the vasorelaxant effect was evident even at doses as low as 0.005 mg/mL (Figure 1A). In support of the literature and animal models, these findings suggested that SDX had an inhibitory effect on constriction mainly via endothelial mechanisms. However, in the endothelium-denuded group, higher concentrations like 0.5 mg/mL were required to see a significant inhibition (Figure 1A). The presence of vasodilation with SDX application in the endothelium-denuded group gave the idea of SDX’s interaction with non-endothelial mechanisms to cause vasodilation. As such, SDX’s inhibitory effect on contraction via non-endothelial mechanisms should be further investigated.

In order to specifically study the role of NO in the mechanism of SDX-induced relaxation, both endothelium-denuded and endothelium-intact IMA rings were pre-incubated with L-NAME. The lack of significant differences between the endothelium-denuded and endothelium-intact groups supported the role of NO in the effect of SDX (Figure 1B). A minimal increase in inhibitory force was observed in both groups, which became more significant after 0.5 mg/mL concentration of SDX application (Figure 1B). Observing the same result in both endothelium-denuded samples and L-NAME precontracted endothelium-intact samples suggested that SDX’s inhibitory force on contraction was mainly caused by the endothelial NO synthase pathway.

This study also involved the Emax and pEC50 values of IMA rings in response to PheE stimulation and cumulative SDX application. The same dose of PheE (3 × 10^−6^ M) resulted in a higher Emax in full-thickness rings when the mechanism leading to vasodilation was disabled with L-NAME pre-incubation (Table 3). Lower values of pEC50 in the endothelium-intact control group indicated reduced sensitivity to PheE. A higher dose of PheE was required to overcome the vasorelaxant contribution of the endothelium. In PheE-induced only IMA rings, physiological results were obtained by reaching higher pEC50 values in groups other than the endothelium-intact control group (Table 3). The vasodilatory effect of the endothelium was abolished by either L-NAME or endothelium destruction. Therefore, half of the maximum contraction response was achieved by a lower concentration. There was also no significant difference in terms of pEC50 between the endothelium-denuded control and the endothelium-denuded L-NAME pre-incubated groups (Table 3). The consequences of PheE induction were the same between these groups. In terms of response to SDX, there was no significant difference in terms of pEC50 between the endothelium-denuded control and the endothelium-denuded L-NAME pre-incubated groups as in the PheE-induced groups (Table 3). However, lower pEC50 values were obtained in endothelium-intact L-NAME pre-incubated and endothelium-denuded groups compared to endothelium-intact control group (Table 3). The result indicated that a higher concentration of agent was needed to reach half of the maximum contraction response. The vasodilatory effect was still present even though inhibition of endothelium-dependent NO pathway and endothelium destruction. This suggested that SDX has an inhibitory effect on arteries not only through the endothelial NO pathway but also through other endothelial or non-endothelial mechanisms. Regarding the documented effect of heparan sulfates on the activation of K_(ATP)_ channels, further investigations are needed to evaluate these in vascular responsivity [50].

IMA is the primary graft of choice for myocardial revascularization to replace diseased coronary vessels [51]. IMA and other small arteries exhibit a tendency to undergo spasms during surgery and in the early postoperative period [52]. This phenomenon can result in significant complications, including graft failure, myocardial ischemia, and, in severe cases, myocardial infarction. Consequently, identifying effective strategies for the prevention and management of arterial spasms is imperative for enhancing surgical outcomes and patient prognosis. Pharmacological interventions constitute one of the primary approaches to mitigate the risk of IMA spasms. Calcium channel blockers, such as diltiazem and verapamil, are extensively utilized owing to their effectiveness in relaxing vascular smooth muscle, thereby preventing spasms [52,53]. Intraoperative topical vasodilators, including nitroglycerin and papaverine, are also applied directly to the graft to induce dilation and reduce the risk of spasms. The remnants of harvested IMA were utilized as arterial samples in this study. The application of SDX resulted in vasodilation by inhibiting arterial contraction through the endothelial NO pathway. It can be concluded that SDX may be beneficial not only in arteriopathy but also in patients undergoing myocardial revascularization. 

The data presented herein represent a basis for future investigations dedicated to better understanding the GCX role in the pathophysiology of arterial insult, with a specific focus on the different anatomic/hemodynamic sites (lower limb, coronary, and cerebrovascular districts).

Although this study contributed significant and valuable data to the literature since it was the first study performed on human tissue, there were some limitations. We conducted the experiment to investigate SDX’s inhibitory effect on arterial contraction in cumulative doses with an ex vivo experiment. Conducting a study with an in vivo design is necessary to observe SDX’s effect more precisely. Moreover, we only used L-NAME to inhibit the NO pathway; more comprehensive and multiple methods may also be used to investigate the SDX’s effect over other vasodilatory mechanisms. The sample size of this study was a limitation. With only 16 patients divided into two groups, a larger sample size and a more diverse population are needed to yield more precise results. Furthermore, the patients in this study presented with various comorbidities, and some were on antihypertensive medication, which could have interfered with the results. While we attempted to exclude certain medications, such as nitroglycerin and calcium channel blockers, prior to surgery and IMA harvesting, other drugs like ACE inhibitors and ARNIs were discontinued only 24 h before surgery due to clinical treatment protocols. Medications such as perindopril and valsartan might have impacted this study, even though their half-lives are less than 24 h. Future studies designed to explore these interferences could provide more comprehensive results. Moreover, regarding the data on the supportive effect of SDX on endothelial cells and the ability of heparan sulfates to modulate vasodilator effects together with its participation in mechanosensing involved in NO production in response to shear stress [54], further studies should be conducted with both morphological and functional studies of SDX-treated individuals’ vascular tissue.

## 5. Conclusions

The primary goal of this study was to determine the effect of SDX on human IMA. SDX creates a concentration-dependent inhibition of arterial contraction. An intact endothelium and NO-mediated pathway are essential for the proper function of that inhibition of contraction. The results of this study demonstrated SDX’s vasodilatory effect besides its known functions, such as anti-inflammatory, antithrombotic, endothelium protective, and profibrinolytic properties.

## 6. Clinical Relevance

Chronic arteriopathy is a progressive disease with a negative impact on quality of life. Chronic arteriopathy encompasses peripheral artery disease, coronary artery disease, and cerebrovascular disease. Although the clinical benefits of SDX have been shown in studies, its mechanism of action on arteries is still unclear, and there are few studies conducted on both animal models and human tissue models. This study represents the first mechanistic investigation conducted on a human artery demonstrating the vasodilatory effect of SDX. To our knowledge, it is also unique for its utilization of human ex vivo harvesting techniques. The use of human tissue causes a more efficient reflection of the mechanism of action over endothelium. The results of this study aid the understanding of SDX’s mechanism of action over arteries. This study also investigates the corresponding pathway of SDX’s effect on arteries. It gives a point of view that more investigation is needed to expand the use of SDX in patients suffering from chronic arteriopathy.

## Figures and Tables

**Figure 1 jcm-13-02332-f001:**
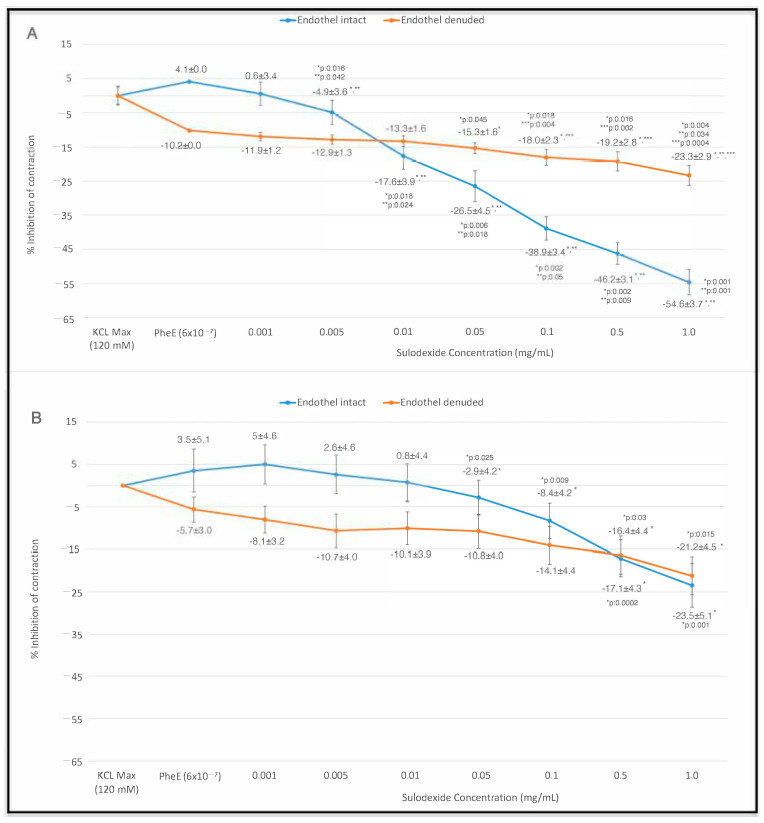
SDX (0.001–1 mg/mL) dose–relaxation (% inhibition of contraction) curves of the PheE stimulated endothelium-intact (n = 8) and endothelium-denuded (n = 8) IMA rings. (**A**) PheE-induced contraction (**B**) PheE-induced contraction after L-NAME (10^−4^ M) pre-incubation. Data are presented as mean ± SEM. The curves depict the variation in contraction force from the baseline. Values that deviate below zero, moving away from the initial point, signify enhanced contraction inhibition. (* *p* < 0.05 within-group comparison, data were compared with PheE-only induced ring; ** *p* < 0.05 within-group comparison, data were compared with previous dose; *** *p* < 0.05 between-group comparison of endothelium-intact and endothelium-denuded rings at a certain dose).

**Table 1 jcm-13-02332-t001:** Demographic data, comorbidities, and medications of the patients whose IMA segments were tested. Patients were divided into two groups according to endothelium integrity. (M, male; F, female; ACE-I, angiotensin-converting enzyme inhibitor; ARNI, angiotensin receptor–neprilysin inhibitor.)

Endothelium Integrity		Intact								Denuded							
Patient Number		1	2	3	4	5	6	7	8	9	10	11	12	13	14	15	16
Age		71	52	68	71	57	68	50	62	70	46	72	62	61	56	71	56
Sex		M	F	M	M	M	M	M	M	M	M	F	M	M	M	M	M
Hypertension		+	−	+	+	−	+	+	+	+	−	+	+	+	+	+	−
Hyperlipidemia		+	−	−	−	−	−	+	−	+	−	−	−	−	+	−	−
Diabetes		+	−	−	+	+	+	+	+	+	+	+	+	−	+	−	+
Use of	Beta blockers	+	+	+	+	+	+	+	+	+	+	+	+	+	+	+	+
	ACE-I	−	−	−	+	−	+	+	−	+	−	−	−	+	−	+	−
	ARNI	+	−	−	−	−	−	−	+	−	−	+	−	−	−	−	−
	Diuretics	+	−	+	+	−	−	−	+	−	−	+	−	−	−	−	−

**Table 2 jcm-13-02332-t002:** The maximum force of contraction (g/100 mg wet tissue weight) recorded from the IMA rings in control and L-NAME pre-incubated groups under basal, KCl-stimulated, and PheE-stimulated conditions. Data are presented as mean ± SEM. Each group was compared for different time points of the protocol: within-group comparisons were made between endothelium-intact and endothelium-denuded rings of the same group (control vs. L-NAME pre-incubated). Between-group comparisons were made between different groups (control vs. L-NAME pre-incubated) of endothelium-intact or endothelium-denuded samples. (* *p* < 0.005 within-group comparison; ** *p* < 0.05 within-group comparison; ^#^
*p* < 0.05 between-group comparison).

	Force of Contraction (g/100 mg Wet Tissue Weight)					
Groups	Control			L-NAME Pre-Incubated		
	Basal Tonus	KCl-Induced	PheE-Induced	Basal Tonus	KCl-Induced	PheE-Induced
Endothelium-intact	3.05 ± 0.37	6.05 ± 0.54	6.33 ± 0.52	3.32 ± 0.12	7.84 ± 0.56 ^#^	7.32 ± 0.57 ^#^
Endothelium-denuded	4.91 ± 0.41 *	8.31 ± 0.68 *	7.55 ± 0.61 **	4.09 ± 0.25	8.32 ± 0.30	7.15 ± 0.4

**Table 3 jcm-13-02332-t003:** Emax and pEC50 values of IMA rings in response to PheE stimulation and cumulative SDX application. Emax, the maximum contraction response, and pEC50, the negative logarithm of the dose to achieve 50% of maximum contraction. * *p* < 0.05 (vs. corresponding control rings); ** *p* < 0.05 (vs. endothelium-intact control rings of corresponding PheE-induced only or SDX-applied group).

	PheE-Induced Only				Response to SDX			
Endothelium Integrity	Intact		Denuded		Intact		Denuded	
Groups	Control	L-NAME Incubated	Control	L-NAME Incubated	Control	L-NAME Incubated	Control	L-NAME Incubated
Emax	102.4 ± 2.8	110.9 ± 2.1	98.4 ± 1.4	101.2 ± 6.5	101.2 ± 8.3	103.2 ± 6.3	97.5 ± 5.8	101.1 ± 7.1
pEC50	6.31 ± 0.16	7.31 ± 0.13 *	7.11 ± 0.13 **	7.05 ± 0.18 **	5.92 ± 0.2	4.80 ± 0.14 *	4.62 ± 0.21 **	4.83 ± 0.19 **

## Data Availability

Additional data is unavailable due to ethical restrictions and privacy.

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
