# Peer review of "Sulodexide Inhibits Arterial Contraction via the Endothelium-Dependent Nitric Oxide Pathway"

_jcm, 2024, doi:10.3390/jcm13082332_

Round 1

Reviewer 1 Report

Comments and Suggestions for Authors

Dear Authors,

Firstly, I would like to congratulate you on your research and paper. Even if the study protocol is rather similar to that of Raffaetto et al. (2019, 10.1016/j.bcp.2019.04.021), it provides ex-vivo insight to other potential vasodilator agents useful in preventing post-CABG graft vasospasm. I consider this manuscript's strong points are its methods section and its results presentation.

However, I consider there are a few aspects that may potentially improve the final form of manuscript:

- Abstract: This could be rephrased to ease legibility "The effect of SDX was more pronounced at endothelium-intact rings except at the lower doses where the percent inhibition of arterial contraction was significantly less in endothelium-denuded segments."

- Materials and Methods: It is stated that there were 16 subjects included - 14 women, 2 men - whereas in Table 1 the frequency is inversely presented M vs. F

- Table 1: Considering this study is originally based on human subjects, I consider it would add clinical value to the paper to present the patients' clinical profile (which may alter the endothelial vasodilatation properties via endothelial dysfunction and lead to confounding effects, especially in the presence of a relatively small study sample) - coronary disease severity, non-coronary artery disease, treatment at the moment of CABG (ACE-I, ARNI, etc.), exposure to diltiazem/nitroglycerine immediately after harvesting

Table 2: Minor change - comparison typo ("comparision") 

Table 2: Overlapping values in the table in the .pdf version, please check

Figure 1: Could you provide an explanation for the paradoxical increased effect of low-doses sulodexide on endothelial-denuded arteries in comparison to endothelial-intact ones? This may be explained by non-endothelial vasodilation mechanisms activated by SDX which should be explained (it is only mentioned in Discussions section)

Limitations section: Although the average values were significantly different in your analyses, I consider it should be clearly emphasized that they were performed on a small sample (n= 8 endo-intact vs. n= 8 endo-denuded). Hence, conclusions such as "SDX’s usage in chronic arteriopathy patients should be promoted and its addition to guidelines should be considered." should be rephrased as there is no current evidence (even based on this study) to justify this clinical decision.

Conclusion: Should only refer to the conclusion that directly arises from the contents of this analysis.

Comments on the Quality of English Language

This manuscript has various typing errors or may require rephrasing of certain expressions:

comparision

"when NO was not in the picture"

IMA and LIMA overlapping abbreviation

Author Response

We deeply appreciate the valuable comments and suggestions provided by the reviewers and the editor. We have thoroughly addressed each point raised during the review process and have made the necessary revisions to our manuscript. We believe that these revisions have significantly improved the manuscript and hope that the changes meet the expectations of the review committee. Thank you once again for the opportunity to revise our work.

  • In abstract section sentences were rephrased for ease of legibility: SDX application resulted in concentration-dependent vasorelaxation in both endothelium-intact (p<0.005) and endothelium-denuded (p<0.05) groups. However, the inhibitory effect of SDX was more pronounced in endothelium-intact rings at higher doses compared to endothelium-denuded rings. Similar inhibition of contraction curves was achieved for both endothelium-intact and denuded rings after L-NAME pre-incubation, suggesting a necessity for NO related endothelial pathway.

  • There was a typing error made during writing of methods section. The study involved 14 male and 2 female patients with a total number of 16 patients.

  • More details about clinical values of patients were given in methods section and in Table 1. All of the patients included in the study were diagnosed with coronary artery disease and planned for elective coronary artery bypass graft surgery. Patients had severe stenosis (70-99%) in two of the major cardiac arteries (left anterior descending, left circumflex, or right coronary arteries.) They underwent double coronary artery bypass graft surgery, which was made by same cardiac team in all patient’s surgery. As anti-hypertensive treatment and cardiac medication, patients were under treatment with metoprolol as beta blocker, Thiazide diuretics, perindopril as angiotensin-converting enzyme inhibitor (ACE-I), and valsartan as angiotensin receptor-neprilysin inhibitor (ARNi). Detail about the medication added to Table 1. Patients were not under treatment with calcium channel blockers. All medication was stopped 24 hours prior to surgery. During the surgery until IMA harvesting, use of Nitroglycerine or calcium channel blockers were avoided. Homeostasis and blood pressure were controlled by the titration of anesthetic agents. After IMA harvest and receiving of remnant segment surgery underwent routinely.

  • Table 2 was modified and corrected.

  • In results section, parts related to figure 1 was rephrased. A paragraph was added to notify the possibility of presence non-endothelial vasodilatory mechanism affected by SDX application. P-values also added to figure 1 and legend of figure 1 was rephrased to give more detail. P values related to within group comparison of data with previous dose were added in figure 1.

  • In discussion(page 10) necessity for further investigation of possible non-endothelial mechanism also supported with a reference. “Regarding the documented effect of heparan sulphates on activation of K(ATP) channels, further investigations are needed to evaluate these in vascular responsivity[50].”

  • Limitations section in discussion were re-analyzed and expanded as:
    “Although our study involved significant and beneficial data to literature, there were some limitations since it was the first study designed on human tissue. We conducted the experiment to investigate SDX’s inhibitory effect on arterial contraction in cumulative doses with ex vivo experiment. Conducting a study with in vivo design is necessary to observe SDX’s effect more precisely. Moreover, we only used L-NAME to inhibit NO pathway, more comprehensive and multiple methods may also be used to investigate the SDX’s effect over other vasodilatory mechanisms. The sample size of the study was a limitation. With only 16 patients divided into two groups, a larger sample size and more diverse population are needed to yield more precise results. Furthermore, the patients in our study presented with various comorbidities and some were on anti-hypertensive medication, which could have interfered with the results. While we attempted to exclude certain medications, such as nitroglycerin and calcium channel blockers, prior to surgery and IMA harvesting, other drugs like ACE inhibitors and ARNIs were discontinued only 24 hours before surgery due to clinical treatment protocols. Medications such as perindopril and valsartan might have impacted the study, even though their half-lives are less than 24 hours. Future studies designed to explore these interferences could provide more comprehensive results. Moreover, regarding the data on the supportive effect of SDX on endothelial cells and the ability of heparan sulfates to modulate vasodilator effects together with its participation in mechanosensing involved in NO production in response to shear stress[54], further studies should be conducted with both morphological and functional studies of SDX-treated individuals’ vascular tissue.”

  • Our study had a small population and focused mainly to demonstrate SDX’s vasodilatory (inhibitory effect over contraction) on arterial samples of human diagnosed with chronic arteriopathy. In order to make the study more comprehensive and broader in future research, bigger sample size with analyze of more variables and pathways are needed.

  • Conclusion and clinical relevance were corrected in accordance with the review.

Reviewer 2 Report

Comments and Suggestions for Authors

Comments on the Quality of English Language

Author Response

We deeply appreciate the valuable comments and suggestions provided by the reviewers and the editor. We have thoroughly addressed each point raised during the review process and have made the necessary revisions to our manuscript. We believe that these revisions have significantly improved the manuscript and hope that the changes meet the expectations of the review committee. Thank you once again for the opportunity to revise our work.

  • Minor editing of English was made in accordance with the review. Abbreviations and spelling were corrected. Changes were made in abstract, results section, discussion and conclusion for ease of legibility.

Reviewer 3 Report

Comments and Suggestions for Authors

Congratulations on a very well written and an important manuscript.

The authors have attempted to look at the effects of SDX on IMA artery.

While mentioning the effects of SDX on endothelium intact or denuded arteries at lower concentration (page 6), the paragraph needs to be modified to provide more clarity. It needs to be specified that the description is meant only for the arteries treated with Phe. Further, the analysis of figure 1 appears to be incorrect. From figure 1, it seems that the % inhibition of contraction for patients treated with Phe. was higher in endothelial intact subgroup, i.e., more potent effect at lower concentration. However, as the SDX concentration increased, the curves crossed each other and the % inhibition of contraction reduced for the endothelium intact subgroup. The authors not only need to highlight this important finding, but also provide a logical and scientific reasoning for this effect.

Based on the above mentioned reason, the conclusion cannot be derived regarding dose related effects of SDX. The relevant caveats need to be mentioned. 

Figure 1, please add confidence interval and/or p-value within the figure for ease of interpretation.

Author Response

We deeply appreciate the valuable comments and suggestions provided by the reviewers and the editor. We have thoroughly addressed each point raised during the review process and have made the necessary revisions to our manuscript. We believe that these revisions have significantly improved the manuscript and hope that the changes meet the expectations of the review committee. Thank you once again for the opportunity to revise our work.

  • Paragraph at page 6 and explanation of Figure 1 in results section was rephrased for ease of legibility. As the Figure 1 shows the inhibition of contraction from the baseline value a negatively sloped curve was obtained from analyzer. Negative values had a meaning of more inhibition in contraction regarding to initial value obtained with only Phe stimulation. For instance, in endothelium-intact group, inhibition at 0.1 mg/ml was 38.9±3.4%. However, a 46±3.1% inhibition was calculated at 0.5 mg/ml. In Figure 1A curve of endothelium-intact group has an increased slope below baseline at higher concentrations (especially after 0.05 mg/ml with a statistically significant difference compared to endothelium-denuded group). In manuscript, an explanatory sentence was placed under Figure 1 in order to clarify the meaning and results explanation of figure 1 was rephrased in page 6 and 7.

  • P-values with significant difference was added to Figure 1. Figure 1 legend was rephrased and corrected for ease of legibility. P values which were referred in Figure 1, refers to comparison of data with PheE-only induced ring; with previous dose; and between endothelium intact-denuded group at certain dose.

Round 2

Reviewer 1 Report

Comments and Suggestions for Authors

My observations have been addressed by the Authors.